

# MicroRNA expression in apical periodontitis and pulpal inflammation: a systematic review

Zainab Jamal Al Gashaamy[1], Tiba Alomar[1], Linah Al-Sinjary[1], Mohammad Wazzan[1], Musab Hamed Saeed[2,3] and Natheer H. Al-Rawi[1]

[1] Oral & Craniofacial Health Sciences, College of Dental Medicine, University of Sharjah, Sharjah, United Arab Emirates
[2] Department of Clinical Science, College of Dentistry, Ajman University, Ajman, United Arab Emirates
[3] Centre of Medical and Bio-allied Health Sciences Research, Ajman University, Ajman, United Arab Emirates

## ABSTRACT

**Background**. The aim of this systematic review is to determine microRNAs (miRs) that are differently expressed between diseased pulpal and periapical tissues.

**Design**. This systematic review used PubMed, Scopus, EBSCO, ProQuest, Cochrane database as well as manual searching to extract studies from January 2012 up to February 2022.

**Results**. A total of 12 studies met the eligibility criteria were included. All selected studies were of case-control type. Twenty-four miRNAs associated with apical periodontitis, 11 were found to be upregulatedand 13 were downregulated. Four out of the 44 miRs associated with pulpal inflammation were upregulated, whereas forty were downregulated. Six miRs, namely hsa-miR-181b, hsa-miR-181c,hsa-miR-455-3p,hsa-miR-128-3p, hsa-miR199a-5p, and hsa-miR-95, exhibited considerable downregulation in both periapical and pulp tissues.

**Conclusion**. MiRs have been investigated for their role in pulpal and periapical biology and may be utilised in diagnostic and therapeutic purposes. Further investigations are required to determine why certain irreversible pulpitis situations progress to apical periodontitis and others do not, based on the various miR expressions. Moreover, clinical and laboratory trials are needed to support this theory.

## INTRODUCTION

Endodontic diseases are one of the most prevalent diseases afflicting dental patients. They are characterized by inflammation of the pulp and periapical tissues and are frequently accompanied by pain and discomfort. Microbial infection of the root canal system and the host's immune response are the primary causes of endodontic diseases, with carious infection being by far the most common cause of pulpal inflammation and necrosis (*Hasselgren & Calev, 1994*). Pulpal and periradicular diseases are microbial in nature (*Kakehashi, Stanley & Fitzgerald, 1965*), in which over 460 bacterial taxa from 100 genera and nine phyla were identified in endodontic infections (*Siqueira Jr & Rôças,*

Corresponding authors
Musab Hamed Saeed,
m.saeed@ajman.ac.ae
Natheer H. Al-Rawi,
nhabdulla@sharjah.ac.ae

*2009a*; *Siqueira Jr & Rôças, 2009b*). In the inflamed pulp, interleukins such as IL-6, IL-8, monocyte chemoattractant protein-1, prostaglandin E2, and inducible nitric oxide synthase are among the inflammatory and immunological mediators (*Fouad & Levin, 2011*). Apical periodontitis is characterized by the deterioration of the periodontal ligament and alveolar bone in the apical region, which is caused by an infected pulp and persistent local immune responses in the periapical region (*Siqueira Jr & Rôças, 2009a*; *Siqueira Jr & Rôças, 2009b*). Despite the prevalence of endodontic disease and the discomfort it causes, the major molecular aspects of its pathogenesis and genetic regulation of pulpal disease remain unknown. Although endodontic infection is common and can be painful, the primary molecular features of its aetiology, as well as the genetic control of pulpal and apical periodontal disease, are poorly understood. MicroRNAs (miRs) are short, non-protein-coding, single-stranded RNA molecules (*Chan et al., 2013*). They consist of 19–24 nucleotides that promote RNA interference by modulating post-transcriptional gene expression (*Chan et al., 2013*; *Lee, Feinbaum & Ambros, 1993*). The human genome contains approximately one thousand miRs (*Al Rawi et al., 2021*). They suppress genes expression by binding to complementary sequences on their target messenger RNA (mRNA), thereby either inhibiting protein translation or initiating messenger RNA cleavage, leading to its degradation (*Ambros, 2004*).

MiRs are essential regulators of gene expression in numerous biological processes, including inflammation, immune response, and osteoclastic bone resorption (*Sonkoly, Ståhle & Pivarcsi, 2008*; *Sugatani & Hruska, 2009*). Multiple diseases, including inflammatory diseases, cancer, developmental abnormalities, cardiovascular disease, and neurodegenerative disorders, have been linked to their dysregulation (*Al Rawi et al., 2021*; *Bartel, 2004*; *Jones et al., 2009*; *Ryan, Robles & Harris, 2010*; *Thum, Catalucci & Bauersachs, 2008*; *Zovoilis et al., 2011*). MiRs are emerging as new biomarkers, prognostic indicators, and therapeutic targets for the diagnosis and treatment of disease (*Chan et al., 2013*). Due to their high sequence conservation across species and tissue specificity, they are suitable biomarkers (*Etheridge et al., 2011*). Recent research has discovered stable miRs in a variety of bodily fluids, including saliva and plasma, allowing for non-invasive miRs profiling (*Mitchell et al., 2008*; *Weber et al., 2010*). Several studies have emphasized the importance of miR in regulating immunity and inflammatory response (*O'Connell et al., 2007*).

The role of miRs in orofacial inflammation, dental tissue development, and the pathophysiology of oral diseases has only recently been determined (*Chan et al., 2013*; *Fouad et al., 2020*). The regulation of toll-like receptor-mediated inflammatory responses, a hallmark of apical periodontitis, has been attributed to miRs (*Galicia et al., 2014*). The first miR study conducted in the field of endodontics revealed that the expression of a large number of miRs significantly differed between healthy and diseased pulps (*Zhong et al., 2012*). Recent miR profiling studies have uncovered a handful of miRs that are specifically associated with pulpal and periapical diseases (*Chan et al., 2013*; *Naqvi et al., 2016*; *Zhong et al., 2012*). Furthermore, three miRs were up-regulated and 33 were down-regulated in inflamed pulps relative to healthy pulps (*Zhong et al., 2012*). *Chan et al. (2013)* performed microarray analysis on inflamed periapical tissues and found that 24 miRs were downregulated. *Xie et al. (2011)* discovered differences in miR expression between inflamed
and normal gingival tissues in a recent study. *Perri et al. (2012)* identified miRs that could connect the molecular pathways between obesity and periodontal inflammation. The goal of this review is to identify how "miR" expression differs between healthy teeth and those with apical periodontitis and pulpal inflammation. Since there were so many miRs under scrutiny but so few studies devoted to individual miRs, it was challenging to develop reliable biomarkers. The merits of our analysis stemmed from the fact that all studies measured miR expression using the same assay technique, which reduced pre-analytical and analytical variability.

## MATERIAL AND METHODS

### Protocol registration

This systematic review was constructed in accordance with the Preferred Reporting Items for Systematic Reviews and Meta-Analyses (PRISMA) guidelines, 2021 (*Page et al., 2021*). It has been registered in accordance with the International Prospective Register of Systematic Reviews (PROSPERO) platform's protocols (ID: CRD42022325848) (*Schiavo, 2019*).

### Focused question

Are "miR"s expressed differently when comparing pulpitis and apical periodontitis?

### Search strategy

Both online and offline browsing was utilized. For this systematic study, five databases were searched: PubMed (https://pubmed.ncbi.nlm.nih.gov), Scopus (https://www.scopus.com/), EBSCO (https://www.ebsco.com), *ProQuest* (https://www.proquest.com/), and the Cochrane database (https://www.cochranelibrary.com/). The search was conducted in February 2022 to discover the differentially expressed miRNA in the periapical and pulpal tissues of patients with periapical or pulpal inflammation. Relevant literature was extracted by additional manual searches of the bibliography.

The following keywords have been used in the search: ("MicroRNA" OR "miRNA" OR "miR") AND ("apical periodontitis" OR "periapical periodontitis "OR "periapical granuloma" OR " apical granuloma" OR " periapical cyst" OR "radicular cyst") AND ("Pulpitis" OR "pulpal inflammation" OR "pulpal pathosis" OR "pulpal necrosis" OR "calcified pulp"). Because the majority of the literature was located in the Cochrane database, the following search terms were employed: MicroRNA or miRNA or miR and apical periodontitis or periapical periodontitis or periapical granuloma or apical granuloma or periapical cyst or radicular cyst and pulpitis or pulpal inflammation or pulpal pathosis or pulpal necrosis or calcified pulp.

### Eligibility criteria

Microarray and/or RT-PCR studies reporting fold changes of down- or up-regulation in gene expression in apical periodontitis or pulpal inflammation and necrosis were included in this study. Only controlled, randomized, case-control, and cohort studies were eligible. Studies in other languages than English were not considered. Review articles, letters, personal comments, literature reviews, systematic reviews and meta-analyses, book

chapters, conference papers, and publications that had been previously published were excluded. Animal studies or those involving humans with additional conditions, such as cancer or viral infections, were also excluded.

## Data extraction

Four investigators (LA, MW, TA, and ZA) who had relevant research background independently examined the papers collected from databases and manual search. This process was accomplished in three steps. The initial step involved the manual removal of duplicates. The second step consisted of reviewing the article titles and removing those that were irrelevant to the discussion topic, followed by reviewing the abstracts of the pertinent titles. A comprehensive search was conducted in the third step, which consisted of reviewing the entire text of the articles that met our inclusion criteria in order to determine which articles qualified. Following a thorough reading, the four reviewers independently assessed the studies' applicability. If the reviewers agreed, the papers were judged to be included in the final step. If there was disagreement, the document would be examined and debated again. In the event that the controversy could not be resolved, an external reviewer (NA)) reviewed the point of contention and eliminated any disagreement.

## Studies quality assessment

The Newcastle-Ottawa scale (NOS) is an easy-to-use tool for evaluating the quality of non-randomized research (*Peterson et al., 2011*). In this systematic review, it was used to assess the quality and risk of bias of the included studies. Utilizing three criteria (selection, comparability, and exposure), the quality of the study was evaluated. Each study was scored on a scale based on stars. A maximum of one star may be awarded for each question in the Selection and Exposure categories. A maximum of two stars may be assigned for comparison purposes. The scientific papers were rated as "Good", "Fair", or "Poor" based on their NOS scores. Studies that received seven or more stars overall were deemed high-quality. Those scoring between four and six are deemed adequate, whilst studies scoring less than four are deemed inadequate.

## Statistical analysis

Using Cohen's kappa coefficient ($\kappa$), the inter-examiner reliability between the four reviewers (LA, MW, TA, and ZA) was determined.

# RESULTS

## Study selection

First, a total of 1,750 articles were identified using electronic databases, and then four non indexed papers were added manually. A total of 367 duplicate documents were omitted. The screening of titles and abstracts resulted in the exclusion of 1,373 papers as irrelevant to this review. The remaining 14 articles were available in full text. Two publications were excluded because they did not adhere to the inclusion criteria, one because its technique differed from ours and the other because it lacked a control group, leaving 12 research for this systematic review. The PRISMA chart also provided the rationale for excluding the

## PRISMA Chart

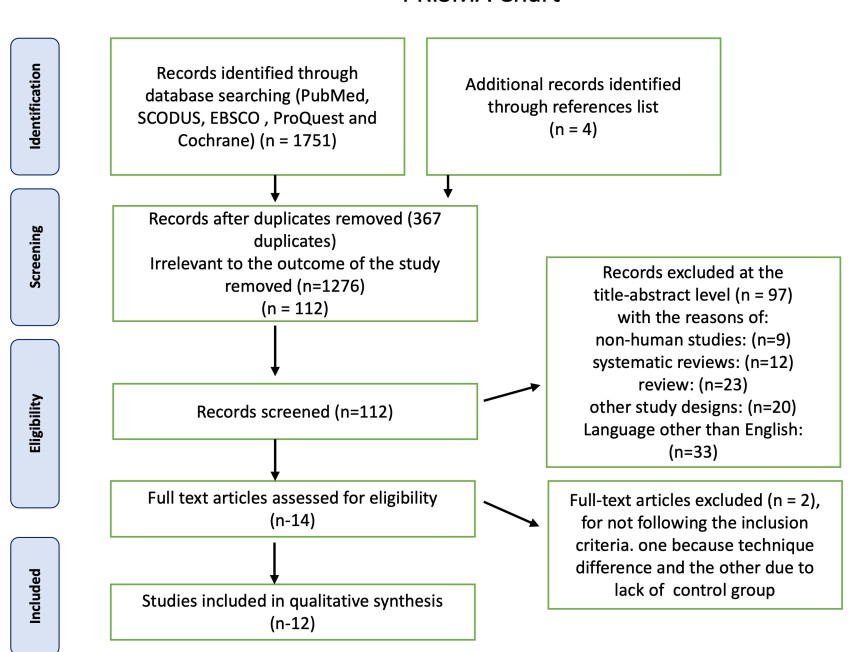

**Figure 1  Summary of the systematic review workflow using a PRISMA chart.**

research Fig. 1. Detailed characteristics of each study are presented in Table 1. For the first and second steps, the inter-examiner reliability between the four reviewers ($\kappa$) was found to be = 0.95, which indicated high reliability.

## Study quality assessment

After evaluating the research, NOS scores ranged between four and eight Table 2. Six articles were deemed excellent with ratings between seven and eight and the other six articles were awarded fair ratings ranging from four to six points. Regarding the subdivisions of the three domains, the risk of bias was determined to be lowest within the selected domain's definition of controls and largest inside the comparison domain.

## Studies characteristics

From the total of twelve studies, eight were done in China (*Huang et al., 2019*; *Lei, Zhang & Xie, 2019*; *Lina et al., 2019*; *Liu et al., 2021*; *Yue et al., 2016*; *Yue et al., 2017*; *Zhang et al., 2019*; *Zhou & Li, 2021*), three in the United States (*Chan et al., 2013*; *Shen et al., 2021*; *Zhong et al., 2012*), and one in Poland (*Brodzikowska et al., 2019*).

A total of 315 samples were collected from the articles, 121 from control groups, including 86 samples of healthy pulp and 35 samples of normal periodontal tissue. The case group consisted of 194 samples, 136 were pulp samples with inflammation and 58 samples with apical periodontitis. One article selected human miR microarrays while the other used next-generation miR-seq analysis to extract miR. Three publications utilized real-time polymerase chain reaction (*Lina et al., 2019*; *Yue et al., 2016*; *Yue et al., 2017*). Seven articles utilized RT-qPCR (reverse transcription quantitative polymerase chain

**Table 1  Characteristic of the included studies.**

| NO. | Authors/Country | Sample | RNA extraction | Study (sample for case and control) | Technique | miRNAs studied | Fold Change\Log Value |
|---|---|---|---|---|---|---|---|
| 1. | *Zhong et al., 2012*, United States | pulp tissue | miRNeasy Mini kit (Qiagen, Valencia, CA) | Healthy Pulp (*n* = 12); Inflammed Pulp (*n* = 18) | Human miRNA Microarrays | 36 were differentially expressed: 3 (miR-150*, miR-584, miR-766) ↑ and 33 ↓. The most significantly repressed expression were miR-664, miR-214, and miR-152. | miR-150* fold change: 2.71, miR-584 fold change: 4.54, miR-766 fold change: 2.17 |
| 2. | *Huang et al., 2019*, China | pulp tissue | kit: TRIzol reagent (Invitrogen, Grand Island, NY, USA) | Healthy Pulp (*n* = 10); Inflammed Pulp (*n* = 10) | quantitative reverse transcription–polymerase chain reaction (qRT-PCR) | hsa-miR-223-3p ↑ , hsa-miR-148a ↓, hsa-miR-146a-5p ↓ and hsa-miR-98-3p ↓. | hsa-miR-223-3p, hsa-miR-148a , hsa-miR-146a-5p and hsa-miR-98-3p fold change >2.0 |
| 3. | *Lei, Zhang & Xie, 2019*, China | pulp tissue | TRIzol reagent (Thermo Fisher, Waltham, MA, USA). | Healthy Pulp (*n* = 4); Inflammed Pulp (*n* = 4) | quantitative reverse transcription–polymerase chain reaction (qRT-PCR) | miR-455-5p ↓ | Fold Change >1.0. |
| 4. | *Zhang et al., 2019*, china | pulp tissues, blood and saliva | phenol chloroform method | Healthy Pulp (*n* = 16); Inflammed Pulp (*n* = 28) | Reverse transcription-quantitative polymerase chain reaction (RT-qPCR) | miR-30b ↓ | ND |
| 5. | *Brodzikowska et al., 2019*, Poland | pulp tissue | RecoverALL Total Nucleic Acid Isolation Kit (Ambion™, Life Technologies, USA). | Healthy Pulp (*n* = 3); Inflammed Pulp (*n* = 3) | quantitative reverse transcription–polymerase chain reaction (qRT-PCR) | miR-410 ↓ | miR-410 fold change >4 |
| 6. | *Zhou & Li, 2021* China | serum & pulp tissue | kit: TRIzol kit (Invitrogen, Carlsbad, CA, USA) | Healthy Pulp (*n* = 34); Inflammed Pulp (*n* = 66) | SYBR Green qRT- PCR | miR-27a-3p ↓ | ND |
| 7. | *Liu et al., 2021*, China | pulp tissue | TRIzol Reagent (Thermo Fisher Scientific, Waltham, MA, USA) | Healthy Pulp (*n* = 7); Inflammed Pulp (*n* = 7) | quantitative reverse transcription–polymerase chain reaction (qRT-PCR) | hsa-mir-30a-5p ↓; hsa-miR-128-3p ↓ | hsa-miR-128-3p: fold change is–1.47; hsa-miR-30a-5p: fold change is–1.22 |
| 8. | *Chan et al., 2013*, United States | periapical tissues and periodental tissue | miRNeasy Mini kit (Qiagen, Valencia, CA) | Apical periodontitis (*n* = 8); normal periodental tissue (*n* = 8) | quantitative reverse transcription–polymerase chain reaction (qRT-PCR) | miR-95 ↓, miR-181a ↓, miR-181b ↓, miR-181c ↓, miR-455-3p ↓, miR-24-1 ↓ and miR-149 ↓ | Log values: miR-95 <−20; miR-181a <−30; miR-181b <−50; miR-181c <−30; miR-455-3p <−20; miR-24-1 <−40; miR-149 <−40 |
| 9. | *Lina et al., 2019*, China | periapical tissue | kit: RNAiso Plus (TaKaRa Bio, Tokyo, Japan) | Apical periodontitis (*n* = 20), normal periodental tissue (*n* = 13) | ABI 7500 real time PCR system with a SYBR® Premix Ex TaqTM II kit (TaKaRa) | miR-146a ↑ | ND |

Al Gashaamy et al. (2023), *PeerJ*, DOI 10.7717/peerj.14949

**Table 1** (*continued*)

| NO. | Authors/Country | Sample | RNA extraction | Study (sample for case and control) | Technique | miRNAs studied | Fold Change\Log Value |
|---|---|---|---|---|---|---|---|
| 10. | *Yue et al., 2016*, China | periodental tissue | Trizol (Life Technologies, Carlsbad, CA) lysis methods | Apical periodontitis ($n = 10$), normal periodental tissue ($n = 6$) | Real-time polymerase chain reaction (PCR) | miR-29b ↑, miR-106b ↑, miR-125b ↑, miR-143 ↑, miR-155 ↑, and miR-198 ↑ | ND |
| 11. | *Yue et al., 2017*, China | periodental tissue | Trizol (Life Technologies, Carlsbad, CA) lysis methods | Apical periodontitis ($n = 10$), normal periodental tissue ($n = 6$) | Real-time polymerase chain reaction (PCR) | miR-335-5p ↓ | ND |
| 12. | *Shen et al., 2021*, United States | periapical tissues and periodental tissue | TRIzol (Invitrogen, Thermo Fisher Scientific, Waltham, MA) | Apical periodontitis ($n = 10$), normal periodental tissue ($n = 2$) | next-generation miRNA-seq analysis (miRNA-seq) | miR-10a-5p ↑, miR-155-5p ↑, miR-7-5p ↑, miR-142-5p ↑, miR-625-3p ↑. miR-891a-5p ↓, miR-514a-3p ↓, miR-128-3p ↓, miR-509-3-5p ↓, miR-199a-5p ↓. | Fold change:miR-10a-5p = 8.44; miR-155-5p =: 4.54; miR-7-5p = 3.67; miR-142-5p = 3.64 = miR-625-3p = 3.50; miR-891a-5p = 0.14; ,miR-514a-3p = 0.19; miR-128-3p =: 0.19; miR-509-3-5p = 0.21; miR-199a-5p = 0.22 |

**Notes.**

\*activated miR150.

ND, Not determined.

**Table 2  Assessment of the quality of included studies using the Newcastle-Ottawa scale (NOS).**

| NO. | Authors/country | selection | Comparability | outcome | NOS score |
|---|---|---|---|---|---|
| 1 | *Zhong et al., 2012*, United States | *** | ** | *** | 8 |
| 2 | *Huang et al., 2019*,  China | ** | ** | *** | 7 |
| 3 | *Lei, Zhang & Xie, 2019*, China | ** | * | *** | 6 |
| 4 | *Zhang et al., 2019*, China | * | * | *** | 5 |
| 5 | *Brodzikowska et al., 2019*, Poland | ** | ** | *** | 7 |
| 6 | *Zhou & Li, 2021*, China | ** | * | ** | 5 |
| 7 | *Liu et al., 2021*, China | ** | ** | *** | 7 |
| 8 | *Chan et al., 2013*, United States | ** | ** | *** | 7 |
| 9 | *Lina et al., 2019*, China | * | * | *** | 5 |
| 10 | *Yue et al., 2016*, China | * | | *** | 4 |
| 11 | *Yue et al., 2017*, China | * | * | ** | 4 |

reaction) (*Brodzikowska et al., 2019*; *Chan et al., 2013*; *Huang et al., 2019*; *Lei, Zhang & Xie, 2019*; *Liu et al., 2021*; *Zhang et al., 2019*; *Zhou & Li, 2021*).

## Study outcome

The 12 studies comprising this systematic review detected a total of 63 miRNA identified by either Microarray or RT-qPCR assay method. Seven publications dealt with pulp inflammation, while the remaining five mentioned the apical periodontitis. Increased expression was detected for 11 of 24 miRs previously linked to apical periodontitis (hsa-miR-7-5p, hsa-miR-10a-5p, hsa-miR-29b, hsa-miR-106b, hsa-miR-125b, hsa-miR-142-5p, hsa-miR-143, hsa-miR146-5p, hsa-miR-155-5p, hsa-miR-198, and hsa-miR-625-3p) and 13 were down-regulated (hsa-miR-24, hsa-miR-95, hsa-miR-128-3p, hsa-miR-149, hsa-miR-181a, hsa-miR-181b, hsa-miR-181c, hsa-miR-199a-5p, hsa-miR-335-5p, hsa-miR-455-3p, hsa-miR-509-3-5p, hsa-miR-514a-3p and hsa-miR-8991a-5p) Fig. 2.

Out of the 44 miRs involved in pulpal inflammation, only four were found to be upregulated (hsa-miR-150*, hsa-miR-223-3p, hsa-miR-584, hsa-miR-766 and hsa-miR-766). Thirty six were downregulated (hsa-miR-24-1*,hsa-miR-27a-3p, hsa-miR-29a*, hsa-miR-29c*, hsa-miR-30a*, hsa-mir-30a-5p, hsa-miR 30b*, hsa-miR-30b, hsa-miR-30c, hsa-miR-30d, hsa-miR-30e, hsa-miR-34b*, hsa-miR-95, hsa-miR-98-3p, hsa-miR-128, hsa-miR-128-3p, hsa-miR-140-3p, hsa-miR-146a-5p, hsa-miR-148a, hsa-miR-152, hsa-miR-181a*, hsa-miR-181b, hsa-miR-181c, hsa-miR-181d, hsa-miR-186, hsa-miR-192, hsa-miR-199a-5p, hsa-miR-214*, hsa-miR-374a, hsa-miR-376c, hsa-miR-410, hsa-miR-455-3p,hsa-miR-629, hsa-miR-664*, hsa-miR- 769-5p and hsa-miR-1308).

Six miRs were found to be down-regulated in both periapical periodontitis and pulpal inflammation (hsa-miR-95, hsa-miR-128-3p , hsa-miR- 181b, hsa-miR-181c, hsa-miR199a-5p, hsa-miR-455-3p).

This article focuses on the reported and quantified differentially expressed miRs in apical periodontitis or pulpal inflammation in which microarray and/or RT-PCR was employed to measure fold changes or down- or up-regulation in gene expression.

**Pupal Inflammation:**
**Down-regulated:**
hsa-miR-24-1*, miR-27a-3p, hsa-miR-29a*, hsa-miR-29c*, hsa-miR-30a*, has mir-30a-5p, hsa-miR-30b*, hsa-Mir-30b, hsa-miR-30c, hsa-miR-30d, hsa-miR-30e , hsa-miR-34b*, hsa-98-3p, hsa-miR-128, hsa-miR-140-3p, ,hsa-miR-186, hsa-miR-192, hsa-miR-214*, hsa-miR-374a, hsa-miR-376c, hsa-miR-410, hsa-miR-629, hsa-miR-664*, hsa-miR-769-5p, hsa-miR-1308.
**Up-regulated:**
hsa-miR-150*, hsa-miR-223-3p,hsa-miR-584 hsa-miR-766,

**Down-regulated in both:**
hsa-miR-95 hsa-miR-128 3p,hsa-miR-181b,hsa-miR-181c, has-miR199a-5p. hsa-miR-455-3p

**Apical Periodontitis:**
**Down-regulated:**
miR-24-1, hsa-miR-149, hsa-miR-181a, hsa-miR-335-5p,hsa-miR509-3-5p, has miR-514a-3p, hsa-miR-891a-5p
**Up-regulated:**
hsa-miR-7-5p, hsa-miR-10a- 5p, hsa- miR- 29b, hsa-mir-106b, hsa- mir-125b, hsa- miR-142-5p, hsa- mir-143, hsa-miR-146-5p, hsa- miR-155-5p, hsa-mir-198, hsa- miR-625-3p

**Figure 2** Summary of the Mirna differently expressed in periapical periodontitis and pulpal inflammation.

## DISCUSSION

This systematic review compiles findings of miR isolated from pulp tissue and the periapical area in the onset of irreversible pulpitis and chronic apical periodontitis. A single miR can influence dozens or even hundreds of distinct mRNAs, making the role of miR in regulating inflammation and the immune response extraordinarily complicated. In contrast, numerous different miRNAs can target a single mRNA. MiRs can also have various functional features depending on the cell type in which they are expressed and can behave as negative gene expression regulators (*Zhong et al., 2012*). The role of noncoding RNAs (ncRNA), both long ncRNAs and short ncRNAs (such as miRs), in physiological and pathologic responses has been investigated (*Lei, Zhang & Xie, 2019*). Due to binding to the 3′-UTR of target mRNAs,the expression of many genes is regulated by posttranscriptional processes, resulting in translational repression or target degradation (*Vasudevan, 2012*). long ncRNAs (LncRNAs) can bind competitively to specific miRs that influence miR-mediated gene silencing downstream (*Khorkova, Hsiao & Wahlestedt, 2015*).These gene interactions can trigger an immune response through inflammatory mediators or cytokines (*Cortez et al., 2019*; *Imamura & Akimitsu, 2014*).

### MiRNA expression in pulpal inflammation

According to published literature, there is increasing evidence that ncRNAs play a critical role in the diagnosis of dental pulp inflammation (*Zhong et al., 2012*; *Huang & Chen, 2018*; *Cortez et al., 2019*).

*Zhong et al. (2012)* found miR-150* was significantly up-regulated in patients with pulpitis. This miR-150* functions as negative regulator of cytokine secretion and collagen biosynthesis, positive regulator of acute inflammatory response, response to thermal and mechanical stimuli.
A miR called miR-584, which specifically inhibits MAPK8, was also found to be up-regulated in pulpitis (*Zhong et al., 2012*). It is thought that this miR is linked to the activation of proapoptotic genes, the JNK cascade, the activation of MyD 88-dependent and independent TLR signaling pathways, the NGF signaling route, and the TLR 1-4 signaling network (*Zhong et al., 2012*).

HSF-1 encodes a heat-shock transcription factor, miR-766 was found to be highly elevated in its pursuit of its target. Heat-shock genes are rapidly activated in transcription following exposure to thermal stress (*Zhong et al., 2012*).

Microarray and quantitative real-time polymerase chain reaction analyses confirmed that miR-233-3p was overexpressed in inflamed pulp samples (*Huang et al., 2019*). In a previous study, it was reported that miR-233-3p was related to pulp inflammation (*Haneklaus et al., 2013*). The involvement of miR-233-3p in odontoblast differentiation suggests a role for it in pulpitis. Markers of odontoblast differentiation and dentin mineralization, such as dentin sialophosphoprotein (DSPP) and dentin matrix protein1 (DMP-1), were considerably upregulated in dental pulp stem cells that overexpresses miR-223 (*Huang et al., 2019*). Several members of the miR-181 family were suppressed. These miRNAs includes miR-181a which is known to regulate IL-6, miR-181b, which regulates CCL8, IL-6, and MMP9 expression, miR-181c, which regulates IL-2 and SOCS1, and miR-181d, which regulates MMP9 expression (*Pichiorri et al., 2008*).

Multiple mechanisms, including inflammatory responses, malignant tumor formation, and epithelial-mesenchymal transition, have been linked to miR-30b through IL-6R signaling pathway (*Graziano et al., 2015*; *Pichiorri et al., 2008*). Similar expression levels of miR30b and IL-6R were detected in plasma and saliva as in pulp tissues, where miR30b was downregulated and IL6R was upregulated (*Zhang et al., 2019*).

Researchers found that miR-410 is downregulated in patients with pulpitis (*Brodzikowska et al., 2019*). According to other study, miR-410 inhibits inflammation *via* the NF-B and HMGB-1 signaling pathway (*Wang et al., 2019*; *Xiong et al., 2017*).

According to a study conducted by *Zhou & Li (2021)* miR-27a-3p was downregulated in patients with pulpitis. Patients with irreversible pulpitis had significantly lower serum and pulp tissue levels of miR-27a-3p than those with reversible pulpitis. Several published research demonstrated that miR-27a-3p targets TLR4 to contribute to the inflammatory response (*Huang & Xu, 2019*; *Zheng et al., 2020*). TLR4 is a member of the TLR family, which is extensively engaged in the inflammatory response (*Hashemi et al., 2018*). A high level of TLR4 expression has been detected in the pulp and periodontium, and it has been implicated in a range of oral pathological diseases (*Stamos et al., 2019*; *Zanini, Meyer & Simon, 2017*).

TLR4 levels were considerably elevated in pulpitis patients' peripheral blood mesenchymal cells (PBMCs) and pulp tissues, and were negatively correlated with miR-27a-3p levels in blood and pulp tissues, respectively.TLR4 has been identified as a direct miR-27a-3p target gene. miR-27a-3p demonstrated equivalent diagnostic accuracy for pulpitis, and its levels may be predictive of irreversible pulpitis in pulpitis patients, as determined by ROC curve analysis (*Zhou & Li, 2021*).

The LncRNA: ADAMTS9-AS2 segment may influence miRNA-related downstream gene silencing by binding competitively to hsa-mir-30a-5p and hsa-miR-128-3p (*Hashemi et al., 2018*). In numerous physiologic and pathological processes, hsa-mir-30a-5p founds to regulates inflammation (*Dong & Wang, 2019*; *Koh et al., 2018*).

LINC00290 is a potential inflammatory biomarker and may also competitively bind to hsa-mir-30a-5p (*Liu et al., 2021*; *MacArthur et al., 2017*). Targeting sirtuin-1,hsa-miR-128-3p can mediate the TNF-a mediated inflammatory response. Both of them are downregulated in samples of pulpitis (*Wu et al., 2020*). SOCS3 and ABCA1, the downstream targets of hsa-mir-30a-5p and hsa-miR-128-3p, may interact with genes in important clusters and hub genes, such as TLR2, CCL2, CXCL10, CD86, TLR10, and IDO1. The fact that half of these genes (TLR2, CCL2, and CXCL10) are leading mirna associated with inflammation suggests a connection between the ceRNA network and pulp inflammation (*Liu et al., 2021*).

MiR-455-5p is a mature form of miR-455 subtype found on the sense strand of chromosome 9q32. MiR-455 is engaged in a range of biological activities *via* the posttranscriptional regulation of particular mRNA. miR-455-5p expression was downregulated in pulpitis samples (*Lei, Zhang & Xie, 2019*).

LncRNA PVT1 has been demonstrated to play a role in Inflammation. Due to its association with miR-455-5p, it may control the progression of pulpitis. In contrast to miR-455-5p, PVT1 was upregulated in pulpitis samples, suggesting a link with pulpitis. Nonetheless, the cause and role remain uncertain (*Lei, Zhang & Xie, 2019*). Two essential miRs (SOCS3 and PLXNC) are significant miR-455-5p target genes. SOCS3 is an important inflammatory suppressor and an inhibitor of the JAK/STAT3 signaling pathway (*Carow & Rottenberg, 2014*). Inflammation is mediated by PLXNC1, an endogenous receptor for the neuronal guidance protein semaphorin 7a. Therefore, both may be suitable targets for preventing damage caused by excessive tissue inflammation (*König et al., 2014*).

PVT1 may compete with miR-455-5p to influence the expression of SOCS3 and PLXNC1. In the instance of pulpitis, the upregulation of PVT1 promotes inflammation and cytokine and chemokine production, while diminishing the expression of miR-455-5p and indirectly raising the expression of SOCS3 and PLXNC1 and the cytokine cascade. SOCS3 and PLXNC1will increase inflammation further by influencing the hub genes (IL-1, IL-6, IL-8, and ICAM1) (*Lei, Zhang & Xie, 2019*).

## MiRNA expression in chronic apical periodontitis (CAP)

CAP is characterized by inflammatory infiltrates that contribute to alveolar bone resorption and tooth loss (*Belibasakis, Rechenberg & Zehnder, 2013*; *da Silva et al., 2012*). The preservation of bone homeostasis is highly dependent on the health of apical bone tissue (*Chen et al., 2015*; *Liu et al., 2017*). Consequently, it is essential to gain a deeper understanding of the processes causing osteoblast- inflammatory responses in CAP.

A growing number of recent investigations have found that aberrant miR expression plays a role in these processes (*Lian et al., 2012*). MiR146a mediates the innate immune response, the inflammatory response, and the antiviral pathway (*Bellon et al., 2009*; *Perry et al., 2008*; *Taganov et al., 2006*). MiR146a5p has been identified as an endotoxin-responsive

gene that regulates many microbial components and pro-inflammatory cytokines (*Taganov et al., 2006*). According to *Lina et al. (2019)* miR146a suppresses the expression of Hey2, IL6, IL1, and TNF in cells. Hey2 has been identified as a miR146a target gene that regulates the expression of miR146a. miR146a and Hey2 produce a mutual negative feedback regulatory loop to modify the inflammatory response in CAP, and the relative expression of both miR146-5p and Hey2 was much higher in the CAP group than in the control group. According to the results,the data, therefore, show that both miR146-5p and Hey2 play a crucial role in CAP pathogenesis (*Motedayyen et al., 2015*).

Multiple disorders have been associated with variations in miR expression. Despite this, the number of studies on miR expression profiles in apical periodontitis is insufficient. *Yue et al. (2016)* reported increased expression of inflammation-associated miRNAs, including miR-29b, 106b, 125b, 143, 155, and 198, in asymptomatic apical periodontitis tissues.

*Yue et al. (2017)* conducted a later investigation which demonstrated that miR-335-5p was significantly down-regulated and had a beneficial effect on the inflammation of human periodontal ligament fibroblasts (HPDLF). In addition, *Yue et al. (2017)* found that miR-335-5p may play a dual role in periapical periodontitis by directly targeting urokinase-type plasminogen activator receptor (uPAR) and receptor activator of nuclear factor kappa B ligand (RANKL), which have strong associations with inflammation and bone destruction, respectively.

*Shen et al. (2021)* used next-generation miRNA-seq analysis of human apical periodontitis (AP) samples to detect differentially expressed miRs in AP. In contrast to past microarray-based miRNA profiling studies (*Shen et al., 2021*). Mir-seq eliminates the necessity for sequence-specific probes, allowing for an unbiased evaluation of the miR expression landscape and the identification of novel miRNAs (*Aldridge & Hadfield, 2012*). When AP tissues were compared to control tissues, 12 miRs were found up-regulated and 94 miRs were found down-regulated, indicating their involvement in gene regulation in the inflammatory and immunological pathways. MiR-10a-5p was the most up-regulated miRNA in AP. These data indicate that variable expression of miR-10a-5p promotes dysregulation of AP-related pathways, including coagulation and macrophage-mediated pathways, which has implications for disease etiology by reducing inflammation and promoting healing (*Shen et al., 2021*).

Several miRNAs from the miR-181 family (miR-181a*, miR-181b, and miR-181c), are down-regulated in diseased periapical tissues relative to healthy periapical tissues. The importance of the miR-181 family in inflammatory diseases is becoming increasingly clear (*Chan et al., 2013*). As this miR is essential in cytokines (IL-2 &6) and chemokines (CCL-8) regulation.

According to previous studies, members of the miR-181 family are significantly down-regulated in inflamed human pulps relative to normal pulps (*Zhong et al., 2012*). MiR-181c, on the other hand, is up-regulated in inflamed gingival tissues (*Xie et al., 2011*).

In addition, miR-24-1*, miR-149, and miR-455-3p were significantly down-regulated in inflamed periapical tissues. The targets of these miRs are also implicated in immunological and inflammatory responses. For instance, MiR-149 suppresses VEGF-a, a protein that acts on endothelial cells to increase vascular permeability and drive cell migration to the site

of inflammation (*Xie et al., 2011*). MiR-455-3p targets both TLR-4 and IL-10, a cytokine produced mostly by monocytes that promotes B-cell survival, proliferation, and antibody production (*Monk, Hutvagner & Arthur, 2010*). MiR 24*-1 target MAPK and downregulate NF-kB pathway and reducing TNFa and IL-6 production (*Xie et al., 2011*).

## Common MiRs in pulpal and periapical lesions

Six miRs, namely hsa-miR-95, hsa-miR-181b, hsa-miR-181c, hsa-miR-128-3p, hsa-miR199a-5p, and hsa-miR-455-3p exhibited considerable differential expression in both periapical and pulp tissues (*Chan et al., 2013*; *Shen et al., 2021*; *Zhong et al., 2012*). All six miRs that were found to be considerably downregulated in inflamed periapical and inflamed pulp tissue when compared to healthy pulp tissue. This indicates that pulpal and periapical disease etiologies have to some extent similar miR regulatory networks. This may be due to the fact that both tissues undergo similar inflammatory processes. Nevertheless, the tissue specificity of miRs is obvious in the distinct types of miRs found in periapical and pulp tissues utilizing microarrays. Multiple signaling pathways and host responses in inflammatory pulp and periapical tissue were shown in Figs. 3A & 3B.

## Limitations of the study

The limited number of relevant articles was one of the key obstacles we faced. In addition, each of the included articles had a small sample size, casting doubt on the validity of the conclusions. In addition, there was a lack of adequate statistical tests such as specificity and sensitivity in some studies, making miR analysis challenging. It was difficult to identify accurate biomarkers due to the enormous number of examined miRs and the small number of research that focused on specific miRs. The miR pathways associated with pulpitis or periapical periodontitis lacked an adequate explanation in some articles.

Our study's strengths and benefits derived from the fact that all studies assessed miR expression using the same assay method, hence decreasing pre-analytical and analytical variability.

Future ramifications are among the most significant advantages of our work. MiRs have been investigated for their role in pulpal and periapical biology and may in future be utilised in diagnostic and therapeutic techniques. As a result of this article, we now have a better understanding of the pathophysiology of pulpitis and apical periodontitis, and we may use this knowledge to develop a therapy that will stop pulpitis from progressing to apical periodontitis.

In the near future, blood or fluid from the pulpal or periapical region can be easily extracted and miR levels can be measured. After evaluating differentially expressed miR, we can draw inferences about prognostic markers and make crucial decisions on patient-centered therapy. After identifying the individual differentially expressed miRs, we can determine which one is involved in the inflammatory process and either suppress or activate them in conjunction with conventional treatments to enhance the healing capacity. This will certainly yield superior results and may be included into contemporary endodontics.

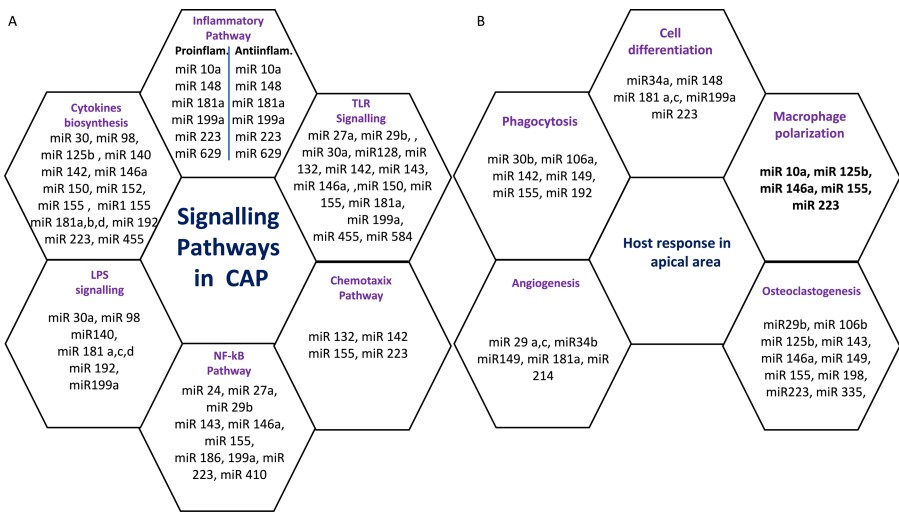

**Figure 3** (A) Common signaling Pathways in pulpal and periapical lesions. (B) Common host response in apical area.

# CONCLUSION

The abundance of miRs was found to be differentially expressed in human CAP tissues and inflamed pulp when compared to normal apical tissues and healthy pulp. Furthermore, six miRs were found to be downregulated in both human CAP tissues and inflamed pulp when compared to normal apical tissues and healthy pulp namely miR-95, miR-128-3p, miR- 181b, miR-181c, miR199a-5p and miR-455-3p. Those six miRs might be the reason why some irreversible pulpitis cases proceed to AP. However, more extensive clinical and laboratory trials are needed to support this theory.

## Funding
The authors received no funding for this work.

## Competing Interests
The authors declare there are no competing interests.

## Author Contributions
- Zainab Jamal Al Gashaamy performed the experiments, analyzed the data, prepared figures and/or tables, and approved the final draft.
- Tiba Alomar performed the experiments, analyzed the data, prepared figures and/or tables, and approved the final draft.
- Linah Al-Sinjary performed the experiments, analyzed the data, prepared figures and/or tables, and approved the final draft.
- Mohammad Wazzan performed the experiments, analyzed the data, prepared figures and/or tables, and approved the final draft.

- Musab Hamed Saeed conceived and designed the experiments, prepared figures and/or tables, authored or reviewed drafts of the article, and approved the final draft.
- Natheer H. Al-Rawi conceived and designed the experiments, performed the experiments, analyzed the data, prepared figures and/or tables, authored or reviewed drafts of the article, and approved the final draft.

## Data Availability

The raw data is available in the Supplemental Files.

## Supplemental Information

Supplemental information for this article can be found online at http://dx.doi.org/10.7717/peerj.14949#supplemental-information.

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
