# Peer review of "MicroRNA expression in apical periodontitis and pulpal inflammation: a systematic review"

_PeerJ, doi:10.7717/peerj.14949_

## Round 0.1 · original submission · Major Revisions

Please revise in detail the comments from each reviewer and submit them at the earliest.

Reviewer 1 ·

Basic reporting

1. The language should be carefully edited and paid attention to sentence grammar and structure so that the text is clear to the readers, such as phrasing makes comprehension difficult, sentences too long or punctuation missing in L35, 131-134, 200-201, 211-217, 219-226. I suggest you have a colleague who is proficient in English and familiar with the subject matter review your manuscript.
2. The language should be expressed more accurately. In L64, “They suppress genes by binding to complementary sequences on their target messenger RNAs” should be “They suppress gene expression by binding to complementary sequences on their target messenger RNAs (mRNAs)”.
3. Both the reference and citation format are not accurate, please check carefully and modify them.
4. There is no corresponding reference cited for PRISMA in L108, PROSPERO in L109, and five databases in L117.
5. Please clarify the Contributions and Competing Interests of all authors in this paper.
6. Figures 1, 2, and 3 have insufficient resolution and need to be improved.

Experimental design

7. It is better to elaborate on why four items were further added manually, and two papers were eliminated in the text in lines 183-186.
8. In L148-149, how these reviewers (LA, MW, TA, ZA, and NA) were independently selected? Are they from the same region or have the relevant research background? Please describe it clearly.

Validity of the findings

9. In the Study Outcome section, whether differentially expressed miRNAs from all studies were identified using the same method and threshold? If not, I suggest that you identify the differentially expressed miRNAs from the raw data with the same method and cutoff.
10. L41: “The identified miRNAs explain why some irreversible pulpitis cases proceed to apical periodontitis and others are not”, how do you make this conclusion? It is not accurate to state this conclusion because you did not verify whether the differential expression of identified miRNAs can cause some irreversible pulpitis cases to proceed to apical periodontitis.
11. Have these differentially expressed miRNAs been reported as biomarkers? What are the biological processes and pathways of upregulated or downregulated miRNAs targeted genes identified in periapical periodontitis, pulpal inflammation, or both? Whether there are miRNA-regulated biological processes and pathways specific or shared in periapical periodontitis or pulpal inflammation? Whether there are biological processes and pathways that may contribute to pulpitis cases proceeding to apical periodontitis? Please add corresponding analysis and make some discussion.

Additional comments

12. Please note the use of the unified number thousands separator in L183 and Figure 1.
13. In L189, 384, and 402, “Table (1)”, “Figure (3)” and “figure3” should be changed to “Table 1” and “Figure 3”.
14. In L206-207 and 262, “Real-time polymerase chain reaction” and “quantitative real-time polymerase chain reaction” should be changed to “RT-PCR” and “qRT-PCR”.
15. In L227, “Six miRNA’s” should be changed to “Six miRNAs”.
16. In L256, “It is thought that this miR is linked to” should be changed to “It is thought that this miRNA is linked to”.
17. Please note the use of microRNA abbreviations, as shown in L64, 79, 84, 85, 90, 113, 268, 339, and 368.
18. Please unify the language in Table 1, such as "Pulp Tissue" and "pulp tissue".
19. In L219-220, do you mean that a total of 33 miRNAs were related to the pulpal inflammation, of which 4 were upregulated and 40 were downregulated? Does it correct?
20. Please use punctuation correctly in L252-254.

Reviewer 2 ·

Basic reporting

1. To improve of basic punctuation, grammar, spacing between words
2. to improve on the standardization of terms and abbreviation used throughout the manuscript
3. Figure 1: Can improve the quality of the figure. There seems to be some distortion in the box at “screening”. Would be good if the reason for exclusion of the papers be more specific (at eligibility section where 2 papers were excluded)
4. Figure 2: Would be good if the words are larger and the circles that overlapped are translucent instead of opaque. It will be more clearer which miRNAs overlapped with pulpal and periapical inflammation.
5. Figure 3: the fonts are too small
6. Figure legends were not provided
7. Table 2: may want to clarify at the footnote of the table what does the asterisk (*) indicate.

Experimental design

1. the aim of the study is not clearly written, hence the research question can be queried
2. to include a stronger justification or problem statement of the study in the introduction
3. To explain further on how they manage disagreement between reviewers

Validity of the findings

1. to provide explanation on the relevance of the findings

Annotated reviews are not available for download in order to protect the identity of reviewers who chose to remain anonymous.

Reviewer 3 ·

Basic reporting

1.The English language should be improved to ensure that an international audience can clearly understand your text.
-Abstract: Twenty-five miRNAs associated to apical periodontitis 12 were found to be upregulatedand 13 were......
Twenty-five miRNAs associated to apical periodontitis, 12 were found to be upregulated and 13 were....

-MicroRNAs (miRNAs)
the abbreviation should be used throughout the manuscript.

-pulpal disease can be irreversible or necrosis?, please specify

-in Search strategy: five databases were consulted: PubMed, Scopus, EBSCO, ProQuest, and the Cochrane database. Please change the word consulted to searched

-please see yellow highlight and annotation in the manuscript information for other corrections

Experimental design

1. Objective:
"The purpose of this review is to determine miRNAs that are differently expressed between diseases pulpal and periapical tissues and their healthy counterparts."

"Are MicroRNAs expressed differently when comparing pulpitis and apical periodontitis ?"

I'm not clear for what exactly do the author want to compare? please be consistent.

2. the word used for search Pulpitis OR pulpal inflammation OR pulpal pathosis OR pulpal necrosis OR calcified pulp, these are totally different terms with different definitions. I'm not sure which one exactly the author intend to look into. It will be very confusing to the reader when these different terms were used interchangeably in the manuscript. It seems like "pulpal inflammation" appeared in all the included studies and this term should be used throughout the manuscript.


3.Statistical analysis
165 Using Cohen’s kappa coefficient (»), the inter-examiner reliability between the four reviewers
166 (LA, MW, TA, and ZA) was determined.
In the results: the inter-examiner reliability of on two reviewers was reported.

Please see the annotation in the manuscript.

Validity of the findings

-please check the number of miRNA reported (yellow highlight and annotation in the PDF file)

Annotated reviews are not available for download in order to protect the identity of reviewers who chose to remain anonymous.

---

## Round 0.2 · Minor Revisions

Please revise and send us asap. Thank you

Reviewer 1 ·

Basic reporting

1. Both reference and citation formats are not correct, please look at the PeerJ journal format for modification. For example, the journal citation should not be numbered and appear after end-of-sentence punctuation. In the reference, the author’s name, publication date, and volume number should be in regular bold font. In addition, there should be no punctuation between the journal title and volume number.
2. In L128, “microRNA (miR)” should be changed to “miR”.
3. Please check and revise the language expression of mature miRNAs, for example, “MIR” should be changed to “miR” in L143, L334, and L393-399.
4. In L243, “figure 2” should be changed to “Figure 2” in regular font.
5. Please unify the font in the whole text, especially in L239-240, L247-253, and L450-451.
6. Please check carefully and revise the punctuation in the full text, such as missing punctuation at the end of the sentence in L387 and L451.
7. In L355, “MiR146a5p” should be changed to “miR146a-5p”.
8. In L449, “Author Contributions” should be in bold font.

Experimental design

no comment

Validity of the findings

no comment

Additional comments

no comment

---

## Round 0.3 · Minor Revisions

Please check the reviewer's comments and submit them at your earliest.

Reviewer 1 ·

Basic reporting

1. Article writing is extremely careless. Authors should carefully double-check whether punctuation font (such as quotation marks on Lines 167, 202, and 219), font color (such as miR on Lines 352, 415, and 416), as well as spaces (such as Lines 70, 166, and 266) and blanks (such as Lines 95, 100, and 136), is consistent throughout the text. In addition, "Good," "Fair," or "Poor" on Line 202 should be changed to "Good", "Fair," or "Poor".
2. Citations of references in this paper should be placed before the period rather than after it.

Experimental design

no comment

Validity of the findings

no comment

Additional comments

no comment

---

## Round 0.4 · accepted · Accept

All comments are addressed to my satisfaction.